# Boosting or inhibiting - how semantic-pragmatic and syntactic cues affect prosodic prominence relations in German

**Stefan Baumann** *, **Janne Lorenzen**

IfL-Phonetik, University of Cologne, Cologne, Germany

* stefan.baumann@uni-koeln.de

## Abstract

In this exploratory study, we investigate the influence of several semantic-pragmatic and syntactic factors on prosodic prominence production in German, namely referential and lexical newness/givenness, grammatical role, and position of a referential target word within a sentence. Especially in terms of the probabilistic distribution of accent status (nuclear, prenuclear, deaccentuation) we find evidence for an additive influence of the discourse-related and syntactic cues, with lexical newness and initial sentence position showing the strongest boosting effects on a target word's prosodic prominence. The relative strength of the initial position is found in nearly all prosodic factors investigated, both discrete (such as the choice of accent type) and gradient (e.g., scaling of the Tonal Center of Gravity and intensity). Nevertheless, the differentiation of prominence relations is information-structurally less important in the beginning of an utterance than near the end: The prominence of the final object *relative* to the surrounding elements, especially the verbal component, is decisive for the interpretation of the sentence. Thus, it seems that a speaker adjusts *locally* important prominence relations (object vs. verb in sentence-final position) in addition to a more *global*, rhythmically determined distribution of prosodic prominences across an utterance.

## Introduction

### Aims and motivation

One of the key prerequisites for successful communication is an appropriate mapping between meaning and form. In spoken language, this mapping implies a connection between different layers of *prominence*, referring to the salience of a syllable or word in comparison with surrounding syllables or words in an utterance. These layers comprise semantic-pragmatic notions such as discourse prominence, predictability, attention-orienting and importance on the one hand, and prosodic prominence, including gradient highlighting via variations in the signal as well as conventionalized linguistic information such as pitch accents or metrical structure, on the other.

The relationship between these two layers is highly complex, since it involves a wide variety of cues on both sides: The meaning side is determined by both lexical properties (e.g., part-of-

**Data Availability Statement:** All data and analysis files are available from the osf database (https://osf.io/s9tmp/).

**Funding:** This work was supported by the German Research Foundation (DFG) as part of the SFB1252 Prominence in Language (Project-ID 281511265),

project A07 Metrical prominence – Scales and structures. The funders had no role in study design, data collection and analysis, decision to publish, or preparation of the manuscript.

**Competing interests:** The authors have declared that no competing interests exist.

speech, word frequency) and the discourse context (e.g., focus, information status), while the prosodic form is not only restricted to phonological categories such as pitch accent types or boundary tones but also covers gradient phonetic parameters like differences in duration or intensity. The meaning-form relationship itself may either be parallel, e.g., increased semantic-pragmatic prominence, as in the case of contrastive focus, is expected to induce increased prosodic prominence, or it may be inverse, e.g., higher semantic-pragmatic prominence in terms of increased accessibility is likely to be marked by less prosodic prominence. The communicative function of the latter relation has been formalized in Aylett & Turk's [1] *Smooth Signal Redundancy Hypothesis*, which suggests that the inverse relation of linguistic predictability and prosodic prominence yields an even distribution of information across the signal making communication more robust.

Consequently, some discourse contextual cues will boost, and some will inhibit the degree of prosodic prominence of an element. Moreover, some factors and their scales are assumed to have a greater impact on an element's prosodic realization than others. The first aim of the paper is thus to provide new insights into possible weighting differences between some semantic-pragmatic and syntactic prominence scales (i.e., referential and lexical newness/givenness, grammatical role, and syntactic position) keeping others (especially lexical factors) constant. By looking at the factors *in combination*, we aim to answer the question whether prominence marking is additive in nature depending on the (type and number of) scales involved.

A second aim is to look at the meaning-form relationship from a decidedly prosodic perspective. That is, we ask which prosodic cues are primarily employed to express specific semantic-pragmatic and syntactic prominence cues and, importantly, what is the effect on the prosodic prominence *relations*, accounting for the fact that prominence is an intrinsically syntagmatic concept.

## State of the art

Previous empirical studies have shown that both production and perception of prosodic prominence are affected by a large number of factors, which have to be taken into account in an investigation of prosodic strength relations (see, e.g., [2–4]). Probably the most comprehensive study on German to date, albeit from a perception side, is [5], testing the influence of 17 linguistic variables on prominence ratings collected in a *Rapid Prosody Transcription* task (see [6]) on read declarative sentences. The choice of variables was based on previous work on West Germanic languages and comprised discrete prosodic variables (e.g., accent types), continuous-valued prosodic variables (e.g., duration of lexically stressed syllables), and non-prosodic variables (e.g., part-of-speech, presence of a focus particle). Results showed that all 17 variables had an impact on perceived prominence. At the same time, they revealed considerable but nevertheless systematic inter-individual variation, which suggests that prominence perception is *both* highly complex and multi-factorial *and* communicatively robust since it allows for a substantial amount of redundancy.

At the group level, an explorative *Random Forest* analysis [7] showed that the discrete prosodic variables accent type and accent position were particularly predictive of perceived prominence, with high and rising pitch accents as well as accents in nuclear position (i.e., the last accent in the phrase) being judged as prominent by the largest numbers of participants (see also [8] for a comparable pattern in American English). This result was in line with a study on (nuclear) accent types in German [9], which established a ranking of accents in terms of their perceived prominence level increasing from low through falling and high to rising accents. Within these categorizations of the F0 contour in the form of accent types, three continuous dimensions have been shown to affect the perceived degree of prominence of a syllable or

word: the actual pitch height (the higher the more prominent, see [10]), the range and slope of the pitch excursion (the larger and steeper the more prominent, see [11,12]) and the alignment of the F0 peak with the metrically strong syllable (the later the more prominent, see [10]). A more recent measure which combines F0 scaling and alignment is the *Tonal Center of Gravity* (TCoG, see [13,14]), which will be described in more detail in the section on Data, annotation and measurements.

Further gradient prosodic cues that have been found to have an influence on prominence in production and perception are segmental duration (the longer the more prominent, e.g., [15–17]) and intensity (the higher the more prominent, especially in high-frequency components; operationalized in various measures calculating overall intensity or spectral balance, e.g., [18,19]). However, in [5], these cues were secondary to the discrete and continuous parameters characterizing the F0 contour. Similarly, [20] in a production study including only continuous phonetic variables found F0 to play the most important role in distinguishing accented from unaccented words and broad from contrastive focus.

Compared to the discrete prosodic parameters, both continuous prosodic and non-prosodic factors only played a subordinate role for the participants' prominence ratings in the study by [5]. Within the group of non-prosodic variables, however, the parameters word frequency and part-of-speech turned out to be particularly important for the listeners. The relevance of these two lexical parameters confirmed previous findings for American English that more frequent words were rated as less prominent [21] and for both German and English that content words are generally perceived (and produced) as more prominent than function words. In turn, nouns and proper names are often rated as more prominent than other types of content words such as verbs and adverbs (see [22] for German; [23] for American English).

Other lexical properties that have been claimed to have an impact on prominence perception as well as on the prominence level of the prosodic output are the semantic weight of a content word and, somewhat related, the (in-)animacy of a referential noun. Bolinger [24] showed for American English that the semantic weight of (in particular) nouns and verbs influences their probability of being accented: the 'heavier' the more prominent, i.e., the more likely to attract the nuclear accent (see also [25] for German). The examples in (1) and (2) differ in the weight of the final verb. In (1), the verb *make* is semantically rather empty, or 'light', so that the nuclear accent is placed on the 'heavier' noun *point* (marked by capital letters). In (2), in comparison, *point* only receives a prenuclear accent (indicated by small capitals) and the nuclear accent falls on the final verb *emphasize*, due to its greater weight, which makes the word less predictable and attracts more attention. Thus, predictability is inversely correlated with both attention and prosodic prominence. Example (3) is added to show the gradience of the relation, in that a semantically empty referring expression like *something* is even less likely to be accented in this context. Importantly, Bolinger states that "it is not necessary for the verb to be fully predictable from the noun; what counts is RELATIVE semantic weight" [24, p. 635].

(1) I have a POINT to make. (2) I have a POINT to EMphasize. [24]

(3) I have something to EMphasize.

Animate entities inherently attract more attention than inanimate ones (e.g., [26]) suggesting a greater effort in marking them. In Russian, for example, an increase in prosodic prominence on animate referents as compared to inanimate ones has been found [27]. At the same time, animate elements are considered more accessible and thus more easily retrievable in discourse than inanimate entities [28,29], which facilitates an occurrence relatively early in an utterance, often as the subject (see, e.g., [30,31] on German). Such a word order effect and the higher degree of accessibility would suggest a realization with lower prosodic prominence to

be more likely. Empirical investigations on the specific relation between (in-)animacy and prosodic prominence are clearly needed, however, in the present study, we restrict ourselves to inanimate referents.

Both grammatical and semantic roles correlate with specific positions in a syntactic clause, which in turn have an impact on a word's prosodic realization. As for grammatical role, subjects often occur first in a sentence (Subject-First preference; see [32] for an overview), while objects occur later, with the last object in a verb phrase usually receiving the nuclear accent rather than the predicate or an external argument (subject) (see, e.g., [33] for English, [25] for German). This makes a (phrase-final) object prosodically the most prominent element in an utterance, at least in broad focus structures. In terms of discourse prominence, subjects are generally considered more accessible and thus more (discourse) prominent than objects (see, e.g., [34,35]). Assuming an inverse relation to prosodic prominence, objects would be expected to be produced as prosodically more prominent than subjects.

Similarly, in terms of the semantic role hierarchy, agents take precedence over non-agents, which lends them higher discourse prominence (see [34]). As for their position in the syntactic clause, agents are often associated with the subject, which in many languages occurs before non-agentive arguments (Agent-First preference, e.g., [34,36]). Prosodically, agents thus often receive a prenuclear accent, while non-agentive roles, e.g., patients, are in many cases marked by a (structurally more prominent) nuclear accent. Moreover, non-agentive subjects more often serve as focus exponents than agentive subjects in German (see [25]), which makes them prosodically prominent as well.

Taken together, it thus seems that the first and last elements in a domain are the most marked ones, with differences at different levels of linguistic description. Both syntactically and semantically, the initial position is often very salient, i.e., prominent on the *discourse* level (e.g., [37]) but not necessarily *prosodically* prominent since it is claimed to be associated with prenuclear accents, which are structurally weaker and less important than nuclear accents, or with no accents at all. Nevertheless, Bolinger [38] proposed a tendency for West Germanic utterances to have one major accent near the beginning, the *accent of power*, and one near the end, the *accent of interest*, constructing a rhythmical frame for the utterance. This tendency was confirmed in a recent production study on sentence topics in German, which found a consistent placement of (prenuclear) accents on sentence-initial referring expressions—in addition to the (nuclear) accent towards the end of the sentence [39].

The next set of prominence cues can only be defined with respect to the previous discourse context, namely focus and information status. There is a large amount of studies, especially on West Germanic languages, which investigated the interrelation of these two levels of meaning with prosody as their main formal indicator. Many studies indicate, e.g., that *new* information is often marked by increased prosodic prominence whereas *given* or *old* information is prosodically attenuated (see, e.g., [40–43]). It has also been shown, however, that this mapping is too simplistic and that intermediate levels of cognitive activation should be assumed, accounting for *accessible* information such as hyponyms and hypernyms, bridging relations etc. (see [44]). Furthermore, two independent levels of givenness have been suggested which potentially influence the prosodic realization of expressions, namely a referential and a lexical level (see the *RefLex* scheme, [45–47]). The following examples illustrate the relevance of such a division:

(4) I went to the dentist yesterday. (5) John got a new car, I'd like to STRANgle the butcher.
 [48] and also BILL bought himself a car.

 r-given r-new l-new l-given
 *The butcher* in (4) is referentially *given* (*r-given*), i.e., coreferential with *the dentist* in the previous sentence, which makes deaccentuation of *butcher* appropriate, despite its lexical

*newness* (*l-new*). The nuclear accent is shifted to the adjacent verb (*strangle*). *A car* in (5) is likely to get deaccented as well but for a different reason: This time, the referent is not the same as a previous one (thus *r-new*) but it is expressed by the same word (thus *l-given*). In sum, both examples exhibit deaccentuation of a referential expression due to *givenness*, which applies either on the referential level as in (4) or on the lexical level as in (5).

Many papers on focus suggest a correlation between pragmatic importance, increasing from broad via narrow to contrastive focus, and prosodic prominence (see, e.g., [49,50] for an overview). Put in a simplified way, the more pragmatically important an element or phrase, or the smaller the number of their contextually salient focus alternatives (following the widely accepted approach of *Alternative Semantics* by [51]), the higher is the probability of a prominent marking of the focus exponent, often by prosodic cues. Focus thus exhibits a parallel increase of discourse prominence and prosodic prominence, i.e., a 'boosting' effect. Although this broad correlation is surely not universal, it is commonly regarded as valid for (West) Germanic languages.

Finally, there are paralinguistic factors such as emotional emphasis or speaking style that influence the prosodic realization of utterances. In general, spontaneous speech has been shown to differ in many respects from read speech—reflecting different levels of control— both at a phonetic and phonological level, but not always displaying consistent differences or similarities (see the collection of papers in [52]). As an example for two relatively controlled speaking styles, clear speech has been found to be manifested in larger pitch range and longer segmental and pause durations than conversational speech (e.g., [53,54] for an overview). Very similar results have been found for TED Talk speakers [4] and other speaking styles that generally involve 'lively' speech, including the increased use of specific accent types (esp. L+H*) ([55] for American English, [56] for German).

To conclude, there is a large number of both prosodic and non-prosodic factors that determine the strength relations between the words carrying multi-dimensional information. Fig 1 summarizes the a) lexical, b) syntactic, c) semantic-pragmatic and d) paralinguistic factors discussed above as scales, which are predicted to decrease (inhibit) or increase (boost) the words' prosodic prominence. Note that the scales are not always continuous–they may consist of discrete levels or binary categories, such as part-of-speech, which could further be divided, e.g., into nouns, proper names, adjectives, verbs and adverbs. In each scale, an increase in prosodic prominence from bottom to top is expected, which means that the meaning-form relationship may either be parallel or inverse, the latter being predicted by the *Smooth Signal Redundancy Hypothesis* [1], as mentioned in the Aims and motivation section above. Previous work has often focused on the influence of a single scale on prosodic prominence production or perception, with the study by [5] taking a more comprehensive approach from the perception side. What has not been done so far is to *combine and weight* the impact of multiple semantic-pragmatic and syntactic prominence cues on prosodic structure and its strength relations in a *production* experiment. This is the aim of the present study. In a reading task involving short stories, we manipulate several syntactic and pragmatic variables (i.e., grammatical role, sentence position, referential givenness and lexical givenness), while keeping other parameters constant (e.g., word frequency, part-of-speech and animacy).

## Research questions and predictions

In the exploratory setup of our production study, we select four semantic-pragmatic and syntactic factors that have been found to influence the level of an expression's prosodic prominence: Referential newness/givenness, lexical newness/givenness, grammatical role and

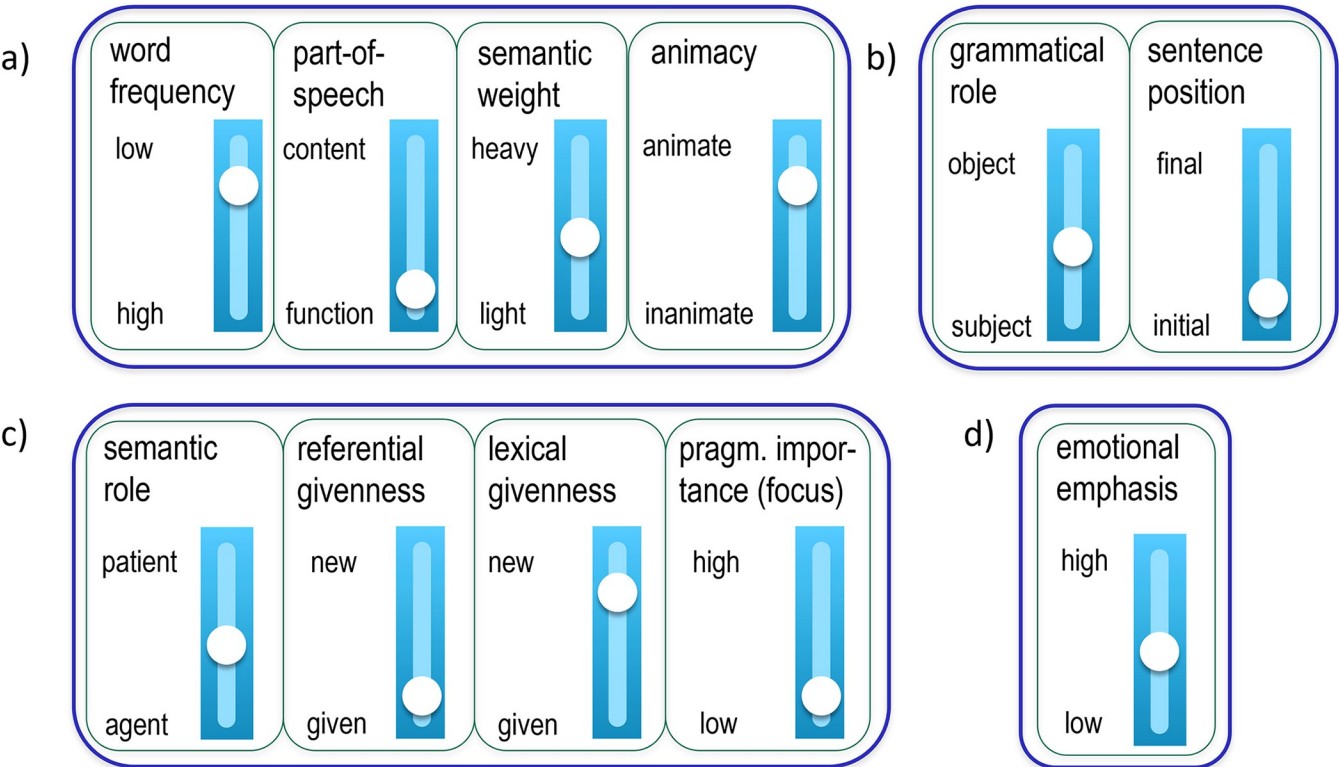

**Fig 1.** Potentially relevant prominence scales for each word: a) lexical, b) syntactic, c) semantic-pragmatic and d) paralinguistic properties; an increase in prosodic prominence from bottom to top is expected.

sentence position. Their impact will be tested on five occurrences of the same target word in a task-oriented setup. Specifically, we consider the following sets of research questions (RQs):

RQ1: Which semantic-pragmatic and syntactic factors boost and which inhibit the prosodic prominence of discourse referents? Which cues have a larger impact than others? Is prominence marking additive in nature?

RQ2: Which prosodic cues are employed to express prominence triggered by semantic-pragmatic and syntactic factors? In particular, what is the effect on the prosodic prominence *relations* within an utterance?

With regard to RQ1, we expect that both referential and lexical *newness*, *object* role, and *medial* position (equivalent to 'final argument') will lead to an increase in the prosodic prominence of a target word (see Table 1 and reading materials in the Materials and methods section for details). Comparing the predicted prominence level of the target words with their prosodic

**Table 1. Predicted (probabilistic) distribution of prominence-boosting (red) and prominence-inhibiting (blue) factors tested in our study, resulting in a ranking of target words according to the number of prominence points assigned to them.**

| | Target1 | Target2 | Target3 | Target4 | Target5 |
|---|---|---|---|---|---|
| Referential givenness | New ↑ | ↓ Given | ↓ Given | New ↑ | ↓ Given |
| Lexical givenness | New ↑ | ↓ Given | ↓ Given | ↓ Given | ↓ Given |
| Grammatical role | Object ↑ | Object ↑ | ↓ Subject | Object ↑ | Object ↑ |
| Sentence position | Medial ↑ | Medial ↑ | ↓ Initial | Medial ↑ | ↓ Initial |
| **Predicted prominence points** | ●●●● | ●●○○ | ○○○○ | ●●●○ | ●○○○ |

realizations will provide information about the *weighting* of the factors tested and their combined effect on prosodic prominence. As a baseline, however, each factor is assumed to have the same weight, i.e., each factor is assigned one prominence point. In fact, we expect an additive effect when the cues combine.

To address RQ2, we will consider several categorical and continuous prosodic correlates of prominence. Most importantly, the placement of the nuclear accent will provide insight into the prosodic strength relations of constituents in an utterance at the syntagmatic level. Prominence-boosting factors will increase a word's probability to be marked by a nuclear accent, while inhibiting factors will decrease this probability.

## Materials and methods

### Reading material

We designed eight short texts in which the same target word occurs in each text in various conditions (see example story in (6)). Keeping the lexical characteristics constant (word frequency, part-of-speech, animacy) allows us to investigate the influence of selected paradigmatic factors on the relative prominence of each target word. Of course, the paradigmatic comparison is to some extent affected by the syntagmatic environment of the word, an aspect which is nevertheless accounted for by the factor sentence position (*initial* vs. *medial*). Target words are common nouns with the same stress pattern (penultimate stress) and equivalent syllable structures (three syllables, stressed open syllable ending on a long vowel, CV:) such as *Limone* (/li'moːne/, 'lemon') or *Praline* (/pʁa'liːne/, 'chocolate candy'). The target words are generally voiced throughout in order to guarantee a continuous pitch contour for our analysis. They were controlled to have comparable frequency classes following the *Wortschatz Leipzig* corpus (https://corpora.uni-leipzig.de/de/res?corpusId=deu_news_2022).

Maria hat im Büro eine Kollegin, mit der sie sehr gut befreundet ist. Fast jeden Tag hat sie ein kleines Geschenk für Maria dabei, das sie ihr in der Mittagspause überreicht. Heute hat die Kollegin **eine Praline (Target1)** mitgebracht. Maria war noch am Arbeiten, als die Kollegin zu ihr an den Schreibtisch kam. Dann hat die Kollegin **die Praline (Target2)** ausgepackt. Sie war von einer edlen Marke und Maria hat sich sehr gefreut. **Die Praline (Target3)** war mit Karamell und Nüssen gefüllt. Als Kind war sie mit Süßigkeiten sehr wählerisch, was auch mit einem Erlebnis auf ihrem achten Geburtstag zusammenhängt. Damals hat Maria **eine Praline (Target4)** gegessen. Sie hat aber nicht gewusst, dass die manchmal mit Alkohol gefüllt sind. Maria hat sich geekelt und danach sehr lange gar keine Schokolade mehr angerührt. Diese Zeiten sind zum Glück vorbei und heute liebt Maria Schokolade aller Art. **Die Praline (Target5)** wird sie gleich als Vorspeise essen.

Maria has a colleague in the office with whom she is very good friends. Almost every day she brings a small gift for Maria, which she presents to her during the lunch break. Today, the colleague brought **a chocolate candy (Target1)**. Maria was still working when the colleague came to her desk. After that, the colleague unpacked **the chocolate candy (Target2)**. It was from a noble brand and Maria was very happy. **The chocolate candy (Target3)** was filled with caramel and nuts. As a child, she was very picky with sweets, which is related to an experience on her eighth birthday. At that time Maria ate **a chocolate candy (Target4)**. But she didn't know that sometimes they are filled with alcohol. Maria was disgusted and didn't touch any chocolate for a very long time afterwards. Fortunately, these times are over and today Maria loves chocolate of all kinds. **The chocolate candy (Target5)** she will eat right away as an appetizer.

For each story, we manipulate four syntactic and discourse-related properties of the target word, which have been found to have an effect on an utterance's prosody. The example story in (6) and Table 1 in section Research questions and predictions illustrate a combination of

these factors. The five mentions of the target word (here: *Praline*) occur as being (i) referentially *given* or *new*, (ii) lexically *given* or *new*, (iii) a *subject* or *object*, and (iv) in an *initial* or *medial* sentence position. The target sentences are further broadly controlled for length and complexity, as well as for the word frequency of the final verb in the 'sentence-medial' condition (the verbs displayed similar frequency classes, taken from the Wortschatz Leipzig corpus; see above). We expect that referential as well as lexical *newness*, *object* function, and *medial* position (which is in fact the final argument in the sentence) will lead to an increase in the prosodic prominence of a target word. If we assign one 'prominence point' (along the lines of [57]) to each of these predicted prominence-boosting factors, we arrive at the accumulated number of points given in the bottom row in Table 1. Comparing the predicted prominence level of the target words with their actual prosodic realizations will provide us with valuable information about the weighting of the factors tested and their contribution to the resulting strength relations.

## Participants and experimental procedure

We collected data from 15 speakers in an interactive reading task (between November 4th and December 6th, 2022). Each speaker produced the same eight stories with five target words in different conditions. Speakers grew up with German as their native language originating from five different federal states of Germany. None of them spoke in a non-standard variety. They were mostly students (at the University of Cologne) at the time of recordings and between the ages of 18 and 31 years (mean: 24 years). Four speakers self-identified as male, eleven as female. Participants provided their written informed consent to participate in this study and received a monetary compensation of eight euros. The Ethics Committee of the German Linguistic Society (DGfS, vote #2020-04-200327) stated that it "has reviewed your application on the basis of the descriptions and information you last submitted on March 26, 2020. Information letters and declarations of consent for conducting behavioral and non-invasive experiments with healthy adult subjects aged 18 to 65 years were submitted and you received a positive vote, as the commission does not currently have any ethical concerns about conducting such a study using procedures established in psycholinguistics. The survey methods do not represent a particular psychological burden for the target age group" (translation from German).

Recordings were conducted in a sound-attenuated booth at the phonetics laboratory in Cologne. Speakers wore a head-mounted AKG C544L condenser microphone. The speech signal was recorded on a computer via *Adobe Audition* at a sampling rate of 44.1 kHz and a bit depth of 16 bit.

Stories were presented in pseudo-randomized order. Each participant first read each story silently to become familiar with it and then read it out loud as part of a listener-directed reading task. A confederate was present during the recording, whom the speakers believed to be another participant. The speaker was instructed to read the stories in such a way that the confederate would be able to memorize the stories and answer three comprehension questions after each story. The confederate answered the questions in written form so that the speaker did not learn about the correctness of the answers at any point during the experiment. Answers were discarded after the experiment. We also presented the speaker with a picture (see Fig 2) for each story to make the experimental design more appealing and reinforce the playful nature of the task. The whole procedure took approximately 30 minutes.

## Data, annotation and measurements

In total, we recorded 600 utterances (15 speakers * 8 stories * 5 target words). Due to hesitations or repairs, we had to exclude 10 utterances. Thus, 590 utterances entered the analysis.

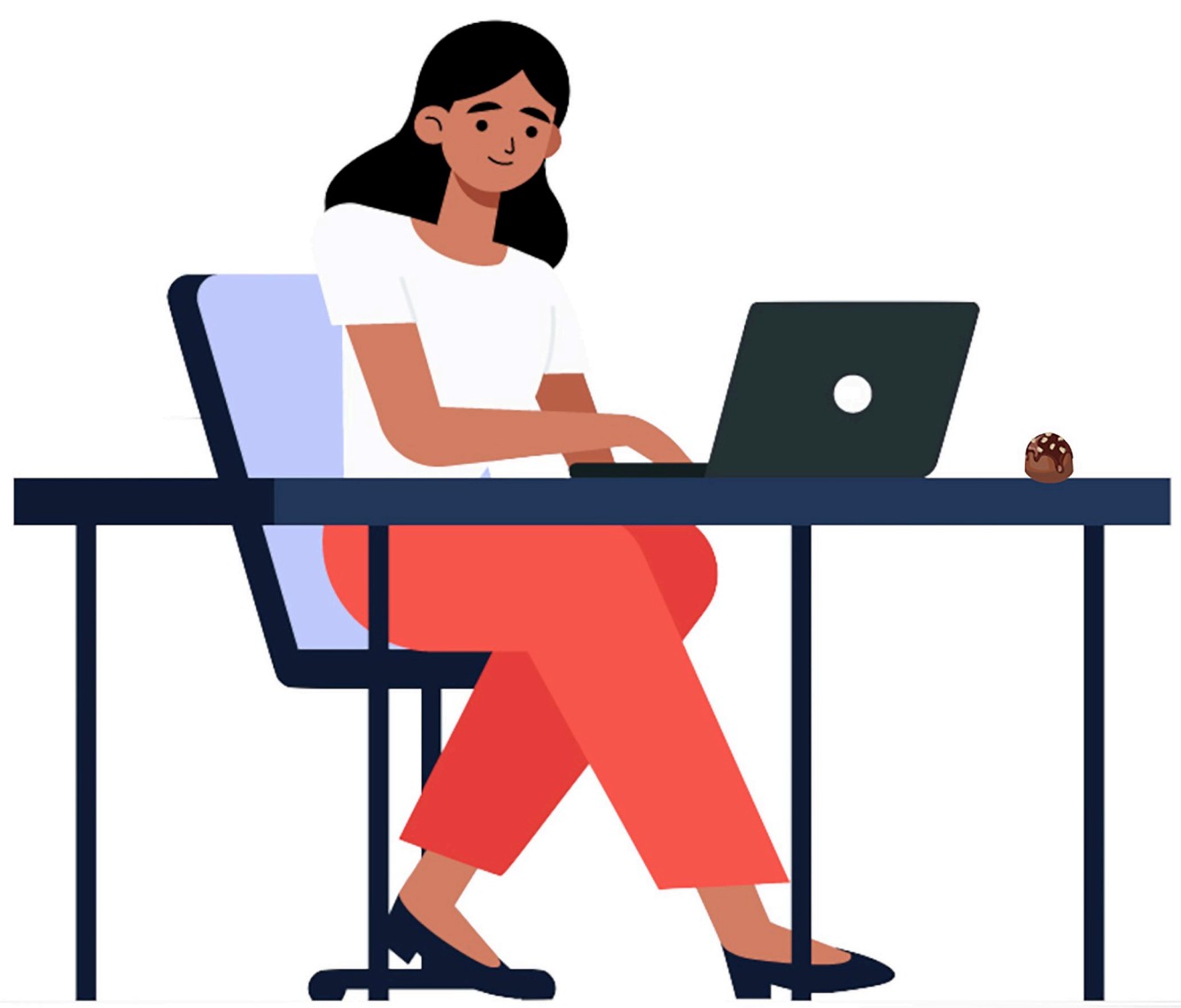

**Fig 2. Picture accompanying the example story in (6) during the reading task.**

The produced target words were analyzed phonologically (accent position and type, following GToBI; [58], perceived prominence and phrase boundary placement according to DIMA; [59]) and acoustically (e.g., measured in terms of duration, intensity, and Tonal Center of Gravity, the latter incorporating the shape of the contour and the alignment of turning points; see [13,14,60]).

All utterances were annotated by two independent transcribers following the DIMA guidelines [59]. Specifically, phrase boundaries were annotated as strong (%) or weak (-) boundaries (agreement rate: 94%). At the tonal level, accentual (starred) and non-accentual tones (turning points, non-starred) were annotated (agreement rate: 71%). Based on these annotations, we determined the accent status (nuclear, prenuclear, deaccented) of each target word. In addition, pitch accents were classified according to the GToBI system ([58]; agreement rate: 64%). For the statistical analysis, we treated pitch accent type as an ordinal scale. Following the findings in [9], we assumed the following ranking of accent types from most to least prominent: L

+H* > L*+H > H* > H+!H* > H+L* >! H* > L* > deaccentuation. Finally, the perceived prominence of words was annotated on a scale of 1 to 3, where 1 indicates weak, often rhythm-dependent prominence, 2 indicates strong prominence that is primarily based on tonal movement and 3 indicates extra-strong prominence, mostly in emphatic realizations (agreement rate: 86%). The absence of any perceived prominence, although not specifically annotated in our dataset, is indicated by a 0 (in DIMA). In cases of disagreement between the transcribers regarding any of the annotated tiers, a consensus was reached with a third expert.

Furthermore, we considered several continuous phonetic parameters. Firstly, we measured alignment and scaling of the Tonal Center of Gravity (TCoG) following Barnes and colleagues [13,14,60]. TCoG measures represent an alternative to traditional measures using turning points and are able to capture both alignment in time and F0 scaling. TCoG alignment constitutes a point in time relative to the beginning of the target word weighted by the F0 contour. Values of TCoG alignment are higher the later the rise and the more 'scooped' the F0 contour are. Values are lower the earlier the rise and the more 'domed' the F0 shape are. TCoG scaling is measured as the weighted average F0 over the target word. Crucially, more sonorous regions contribute more to the value of TCoG scaling [60]. In order to get rid of participant differences in these values, we converted them to a semitone scale relative to a speaker-specific baseline. This baseline was determined by extracting the F0 values of the last low tone (i.e., non-accentual L or accentual L*) annotated in each utterance and taking their median for each speaker.

Furthermore, we measured the duration in milliseconds and the mean intensity in decibels of the stressed syllable of the target word. For the statistical analysis, we excluded phrase-final instances of the target words from the duration measures to control for potential effects of final lengthening.

## Statistics

We ran Bayesian mixed-effects regression models using the *brms* (version 2.20.1, [61]) package as an interface to the *Stan* modeling language (version 2.26.2, [62]) in *R* (version 4.3.1, [63]). We built one model for each prosodic parameter, with the parameter in question as the dependent variable. For categorical dependent variables, we used binary logistic or cumulative models, for continuous variables we ran linear regression models.

Each model included four predictor variables: Position, grammatical role, referential givenness and lexical givenness. The reference levels for these predictors were *initial* (vs. *medial*), *subject* (vs. *object*) and *given* (vs. *new*) for both levels of givenness. To account for inter-speaker and inter-item variability, we included random intercepts for item and speaker and random slopes for the by-item and by-speaker effects of all predictors. Four sampling chains were run for 4,000 iterations each with a warm-up period of 2,000 iterations, yielding a total of 8,000 posterior samples per model. For the regression coefficients, we specified weakly informative, normally distributed priors with a mean of zero and a standard deviation of ten for the linear regression models and a mean of zero and a standard deviation of one for the cumulative models. We used default priors supplied by *brms* for the remaining model parameters.

We compared the posterior estimates of each variable to the reference levels. Positive values in these comparisons provide evidence for the predictions specified in the Research questions and predictions section. For each model, we report the regression coefficient β and 90% credible intervals (CIs) under the posterior distributions as well as the posterior probability that β is larger than zero (Pr(β > 0)). When the CIs do not include zero and the posterior probability Pr(β>0) is larger than 0.95, we take this as compelling evidence in favor of our predictions.

For data processing and plotting, we used the *tidyverse* (version 2.0.0, [64]) and *ggdist* (version 3.3.0, [65]) packages. For each parameter, we plot the distribution of the measured data

by target word number and the posterior estimates from the regression models for each predictor variable. In the model plots, the colored shapes represent the distributions of the posterior estimates, the dot is the mean, the thin horizontal lines indicate 90% CIs and the thick lines 66% CIs. The vertical dashed line marks the position where the estimate β equals zero, i.e., where there is no difference between the levels that are being compared. When the thin black line does not overlap with the vertical zero line, the criterion that zero is not included in the 90% CI is fulfilled. Data and code for analysis are available on the OSF repository under the following link: https://osf.io/s9tmp/.

## Results

In the following sections, we examine the influence and ranking of the four semantic-pragmatic and syntactic predictors *referential givenness*, *lexical givenness*, *sentence position* and *grammatical role* on specific prosodic parameters, which in turn signal differences in the level of (prosodic) prominence. The following two sections present the results regarding a) the discrete prosodic parameters and b) the continuous prosodic parameters.

### Discrete prosodic parameters

The first and central parameter we investigate is accent status. Target1 is usually produced with a nuclear accent (in 96% of the cases), as is Target4 (75%, see Fig 3, left). Target2 and Target3 are predominantly produced with prenuclear accents, in 73% and 83% of cases, respectively. Target5 is almost equally likely to receive a nuclear or prenuclear accent (42% and 57%, respectively). Thus, the probabilistic distribution of the target words' accent status generally confirms our predicted prominence ranking (see Table 1) with one exception: Target2 and Target5 switched their ranks, since Target5 is produced with more nuclear accents than Target2, which runs counter to our expectations.

Since deaccentuation occurs in only 12 target words, we treat accent status as a binary variable with the levels *nuclear* and *prenuclear/deaccented* and run a binary logistic regression model. A further justification for this decision stems from the fact that in annotation,

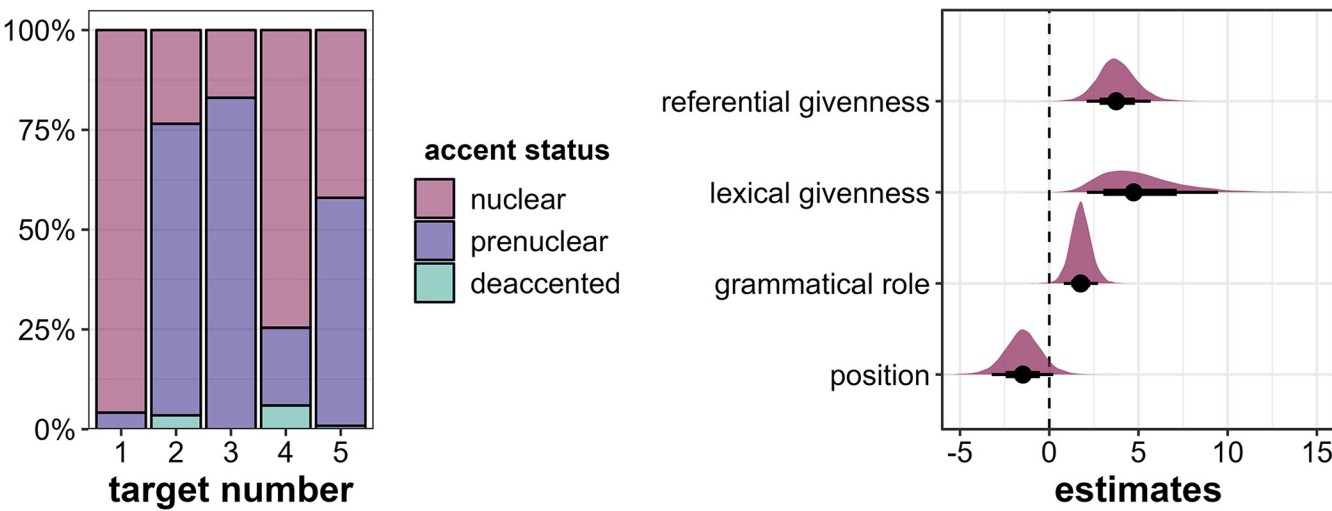

**Fig 3. Results for accent status.** Left: Distribution of accent status across the five target word positions in percent. Right: Posterior estimates of referential givenness, lexical givenness, grammatical role and position for the effects of accent status as predicted by the model, means, 66% (thick horizontal lines) and 90% credible intervals (thin lines).

prenuclear and deaccented targets are often difficult to distinguish, while it is usually a straightforward task to determine the location of the nuclear accent.

The Bayesian mixed-effects regression model confirms that there are compelling effects of referential givenness (β = 3.81, CI = [2.1; 5.69], Pr(β>0) = 1), lexical givenness (β = 5.13, CI = [2.11; 9.46], Pr(β>0) = 1) and grammatical role (β = 1.76, CI = [0.82; 2.74], Pr(β>0) = 1) in the direction we expected (see Fig 3, right). That is, referentially and lexically *new objects* are more likely to be realized with a nuclear accent than referentially and lexically *given subjects*. However, there is some evidence that the *initial* position is more prominent than the *medial* position, which we did not anticipate (β = -1.49, CI = [-3.22; 0.23], Pr(β>0) = 0.07).

As evident from Fig 4, we observe some inter-item variability, especially in Target2 and Target4. While Target2 is often the least likely to receive a nuclear accent (between 0% and 25% nuclear accents), i.e., the least prominent target word in a story, it is more likely to receive a nuclear accent in the stories containing *Banane* ('banana', ca. 50% nuclear accents) and *Gardine* ('curtain', ca. 75% nuclear accents). Target4 is usually the second most prominent target word following Target1, which is expected since the two conditions differ only in one factor, namely lexical givenness. However, in the stories related to *Banane* ('banana') and *Garage* ('garage') Target4 is equally prominent as Target1 suggesting that lexical givenness does not inhibit the prominence of Target4 in these stories. In the stories containing *Gardine* ('curtain') and *Kombüse* ('galley'), Target4 is relatively *less* prominent than Target5 and less prominent than Target2 in the story containing *Gardine*. This could indicate that referential *newness* in Target4 does not boost prominence as much as in the other stories or that lexical and

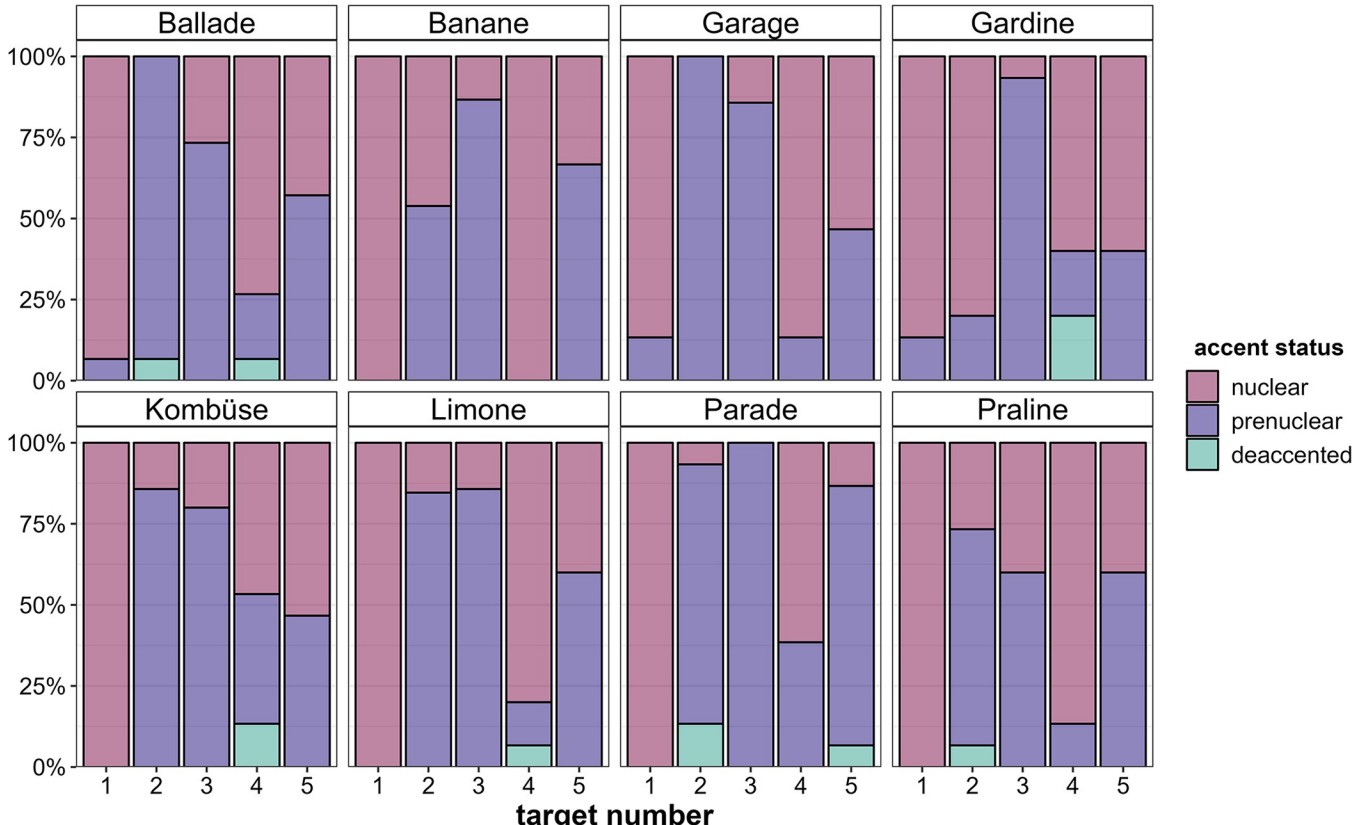

**Fig 4. Distribution of accent status in percent across the five target word positions facetted by item/story.**

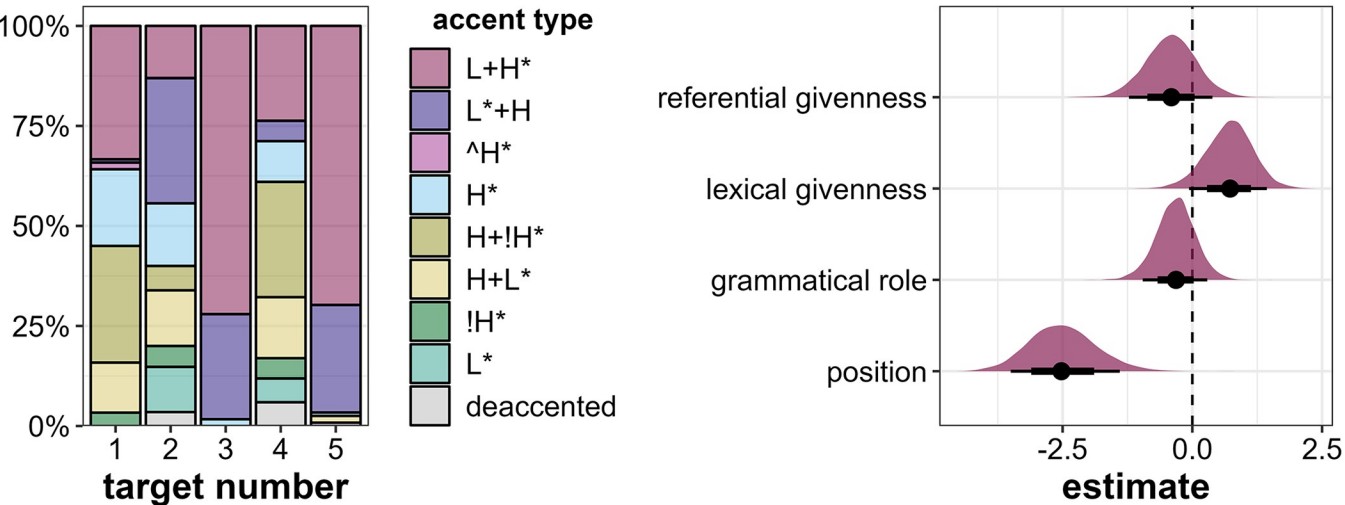

**Fig 5. Results for accent type.** Left: Distribution of accent types across the five target word positions in percent. Right: Posterior estimates for the effects of referential givenness, lexical givenness, grammatical role and position on accent type as predicted by the model, means, 66% (thick horizontal lines) and 90% credible intervals (thin lines).

referential givenness in Target2 in the *Gardine* story does not inhibit prominence as much. There are also some consistencies across stories in that Target1 is always the most prominent target word, and in the initial position, Target5 is usually more prominent than Target3. Both of these results are in line with our expectations for the specific cue combinations of these conditions.

In terms of pitch accent type, Target3 and Target5 (both in initial position) are realized very similarly, mostly with the very prominent L+H* (ca. 70%) and L*+H (ca. 25%) accents (see Fig 5, left). In the medial position, there is more variability regarding the produced accent types, especially in Target2 and Target4. In Target1, there is a larger proportion of L+H* accents (33%) as compared to Target2 (13%) and Target4 (24%). Target1 and Target4 are also often realized with falling H+!H* accents (both 29%). Target2 often receives L*+H accents (31%).

According to our regression model, there is some evidence that lexically *new* target words are produced with more prominent accent types than *given* targets, however, this evidence is not compelling according to the criteria specified in the Statistics section ($\beta = 0.71$, CI = [-0.06; 1.44], Pr($\beta > 0$) = 0.94, see also Fig 5, right). Referential givenness and grammatical role do not seem to have an influence on accent type choice in the expected direction (referential givenness: $\beta = -0.41$, CI = [-1.22; 0.39], Pr($\beta > 0$) = 0.2), grammatical role: $\beta = -0.32$, CI = [-0.96; 0.29], Pr($\beta > 0$) = 0.19). However, position is clearly a decisive factor for accent type choice, in that the *initial* position attracts especially strong accents ($\beta = -2.5$, CI = [-3.5; -1.4], Pr($\beta > 0$) = 0.00).

As evident from Fig 6 (left), most target words are produced with a full pitch accent corresponding to the DIMA prominence label 2. However, there are some weaker productions of target words corresponding to DIMA levels 0 and 1, especially in Target2 and Target4, both occurring in *medial* position.

The regression model suggests that there is compelling evidence to assume that lexically *new* target words are produced more prominently than lexically *given* ones ($\beta = 1.52$, CI = [0.06; 2.84], Pr($\beta > 0$) = 0.95, see Fig 6, right) and some evidence that referentially *new* target words are produced more prominently than referentially *given* ones ($\beta = 0.55$, CI = [-0.48;

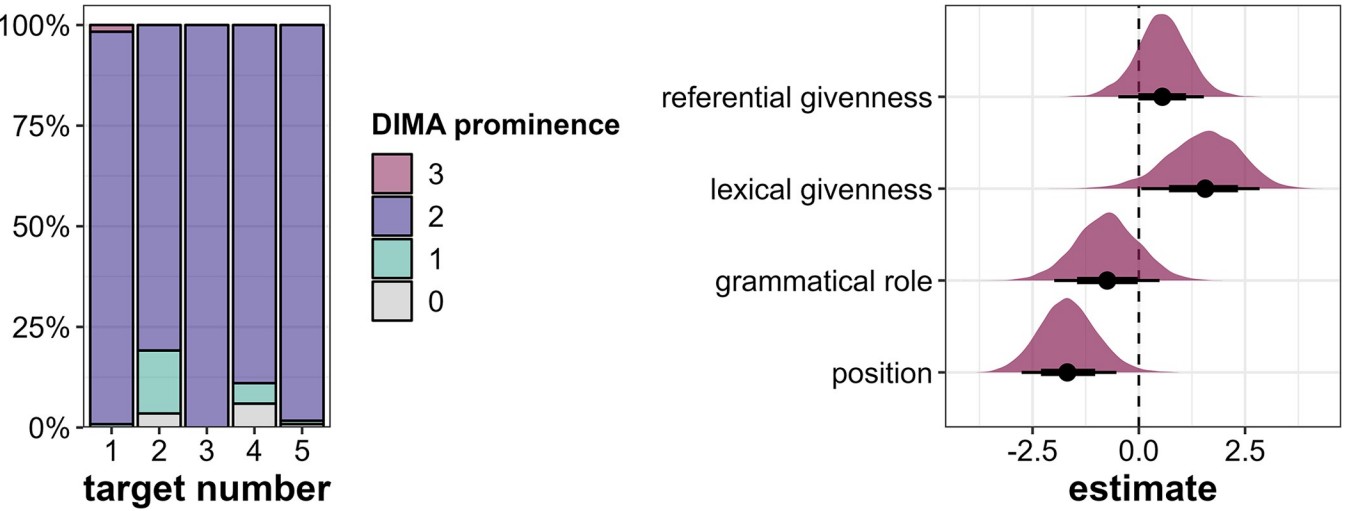

**Fig 6. Results for DIMA prominence scores.** Left: Distribution of prominence scores across the five target word positions in percent. Right: Posterior estimates for the effects of referential givenness, lexical givenness, grammatical role and position on prominence scores as predicted by the model, means, 66% (thick horizontal lines) and 90% credible intervals (thin lines).

1.53], Pr(β>0) = 0.83). However, contrary to our expectations, there is some indication that grammatical *subjects* are perceptually more prominent than *objects* (β = -0.75, CI = [-1.99; 0.49], Pr(β>0) = 0.16). Again, position has a large unpredicted influence in that *initial* target words are perceived as more prominent than *medial* target words (β = -1.66, CI = [-2.76; -0.53], Pr(β>0) = 0.01).

Phrase boundaries occur only after Target3 and Target5, i.e., after the target words in the *initial* sentence position (see Fig 7, left). This is to be expected, as the *medial* target words occur in the last slot before the utterance-final verb, where a phrase boundary would be highly marked. The lack of boundaries after *medial* target words can thus not be interpreted as a lack of prominence. Therefore, phrase boundaries as prominence markers can only be investigated

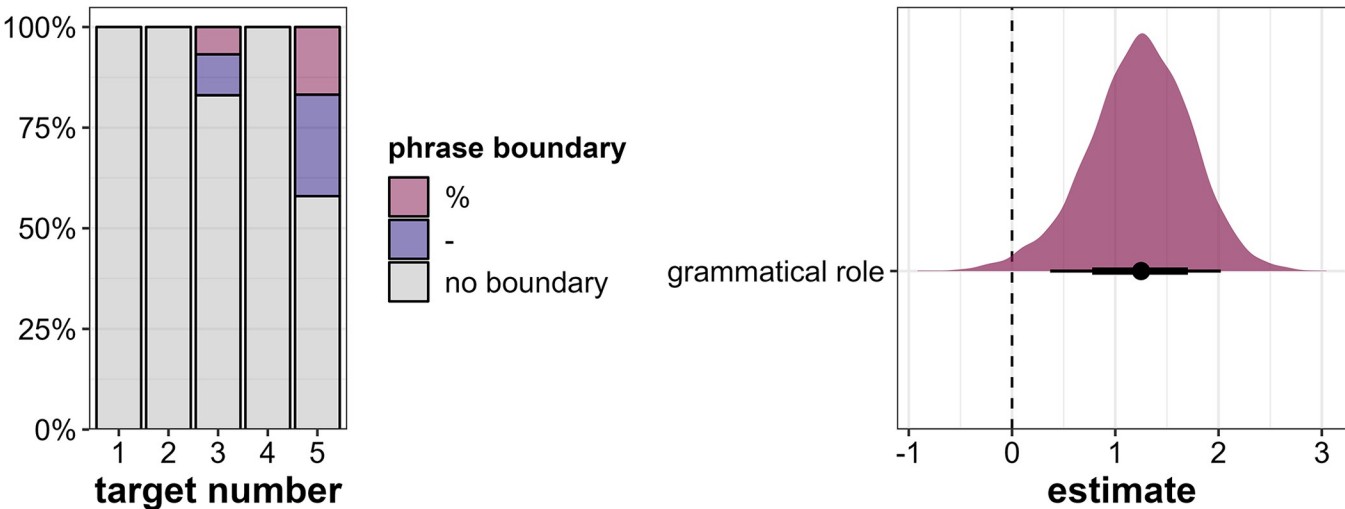

**Fig 7. Results for phrase boundary placement.** Left: Distribution of phrase boundaries across the five target word positions in percent. Right: Posterior estimates for the effects of grammatical role on prominence scores as predicted by the model run on a subset of the data including only Target3 and Target5, means, 66% (thick horizontal lines) and 90% credible intervals (thin lines).

in the *initial* position and the corresponding regression model is run on a subset including only Target3 and Target5. As a consequence, we can include only one predictor, namely grammatical role, since Target3 and Target5 do not differ in any other predictors. The model provides evidence that *objects* are more likely to be followed by more and stronger phrase boundaries than *subjects* (β = 1.23, CI = [0.37; 2.02], Pr(β>0) = 0.99, see Fig 7, right).

### Gradient prosodic parameters

The Tonal Center of Gravity is located farthest from the beginning of the target word in Target3 (mean = 298 ms, sd = 60 ms) and Target5 (mean = 320 ms, sd = 59 ms), i.e., the *initial* target words (see Fig 8, left). This is due to the large number of rising pitch accents in this position. Target5 exhibits on average slightly higher values, i.e., later peak alignment, than Target3. In the *medial* position, Target2 has the latest peak alignment (mean = 221 ms, sd = 52 ms), followed by Target1 (mean = 210 ms, sd = 50 ms) and Target4 (mean = 191 ms, sd = 47 ms), although these differences appear to be relatively small.

The regression model confirms that position is the most important factor in determining alignment in that TCoG in *initial* target words is later aligned than in *medial* words (β = -11.0, CI = [-28.06; 6.42], Pr(β>0) = 0.15, see Fig 8, right). Unexpectedly, referentially *given* target words also appear to exhibit later TCoG alignment than *new* target words (β = -9.31, CI = [-24.31; 5.96], Pr(β>0) = 0.16). However, lexical givenness behaves as expected, with lexically *new* target words exhibiting a later TCoG than *given* words, although this evidence is not compelling according to our criteria (β = 7.08, CI = [-6.91; 20.18], Pr(β>0) = 0.8). Grammatical role barely affects TCoG alignment (β = 3.63, CI = [-11.22; 17.57], Pr(β>0) = 0.67).

Similarly to TCoG alignment, TCoG scaling also reflects the large number of rising accents in the *initial* position with the highest mean values occurring in Target5 (mean = 6.0 st, sd = 2.5 st) and Target3 (mean = 5.8 st, sd = 2.3 st, see Fig 9, left). In the *medial* position, Target1 has the highest mean value (4.0 st, sd = 2.9 st), followed by Target4 (mean = 3.3 st, sd = 2.4 st) and Target2 (mean = 3.1 st, sd = 1.9 st), although these differences are again very small.

The regression model provides support for position as the most important factor in determining TCoG scaling with the *initial* target words reliably exhibiting higher values (β = -2.83,

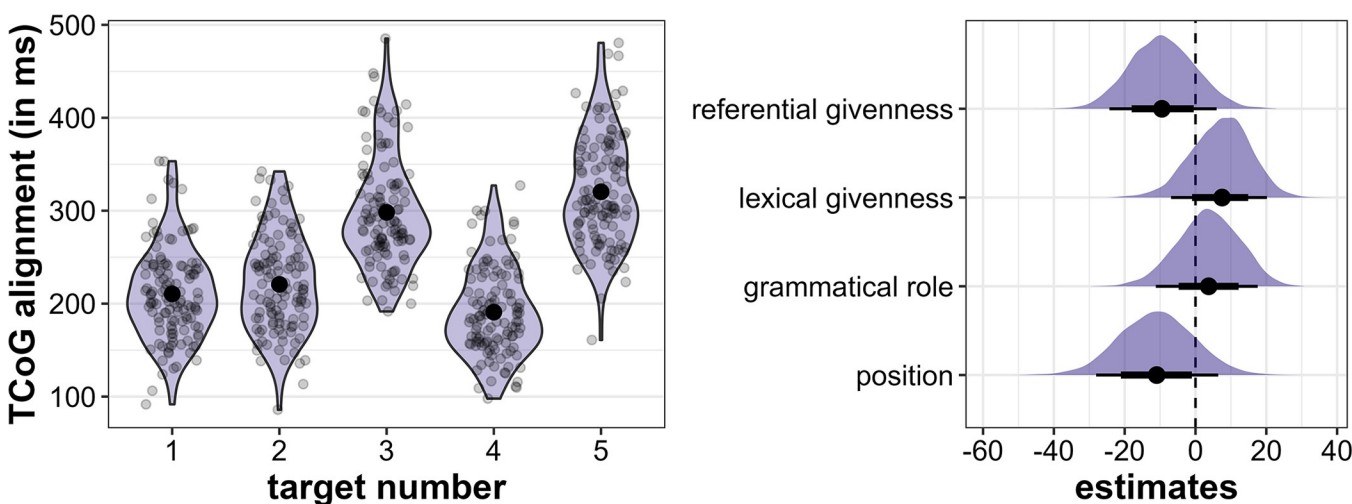

**Fig 8. Results for TCoG alignment.** Left: Distribution of TCoG alignment (in ms) across the five target word positions. Right: Posterior estimates for the effects of referential givenness, lexical givenness, grammatical role and position on TCoG alignment as predicted by the model, means, 66% (thick horizontal lines) and 90% credible intervals (thin lines).

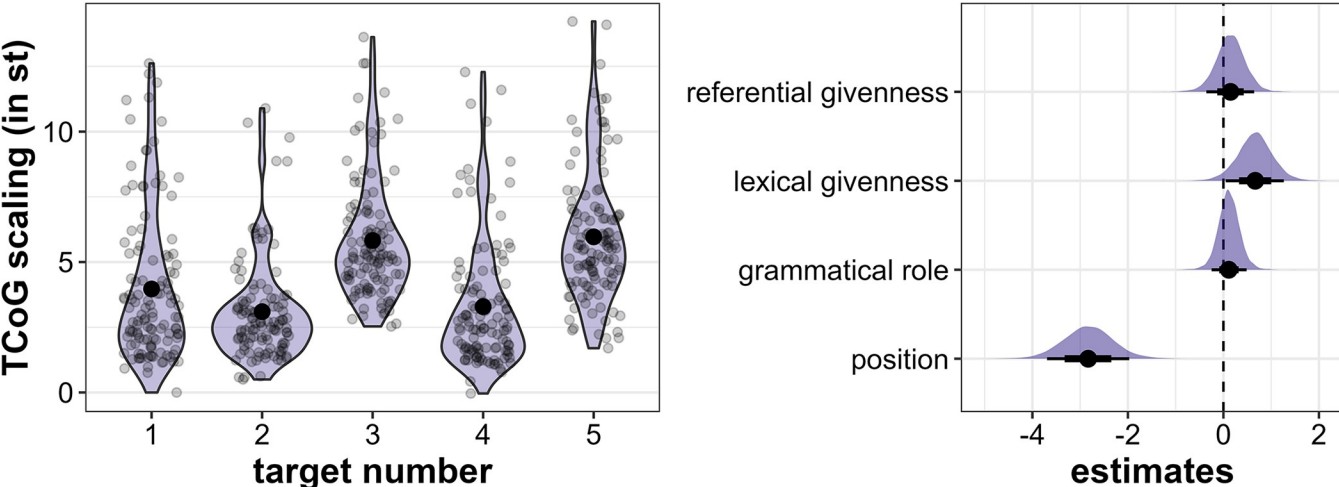

**Fig 9. Results for TCoG scaling.** Left: Distribution of TCoG scaling (in st) across the five target word positions. Right: Posterior estimates for the effects of referential givenness, lexical givenness, grammatical role and position on TCoG scaling as predicted by the model, means, 66% (thick horizontal lines) and 90% credible intervals (thin lines).

CI = [-3.69; -1.97], Pr($\beta$>0) = 0.00, see Fig 9, right). In addition, there is compelling evidence that lexically *new* target words are produced with higher TCoG scaling ($\beta$ = 0.66, CI = [0.05; 1.27], Pr($\beta$>0) = 0.96). Although referentially *given* words and words in *object* role also appear to be higher in scaling, evidence to support these tendencies is much weaker (referential givenness: $\beta$ = 0.15, CI = [-0.36; 0.64], Pr($\beta$>0) = 0.7, grammatical role: $\beta$ = 0.12, CI = [-0.25; 0.49], Pr($\beta$>0) = 0.71).

In terms of syllable duration, the accented syllable in Target1 is on average longest (mean = 192 ms, sd = 35 ms), followed by Target5 (mean = 190 ms, sd = 35 ms) and Target3 (mean = 183 ms, sd = 34 ms), and Target4 (mean = 173 ms, sd = 33 ms) and Target2 (mean = 171 ms, sd = 35 ms) are the shortest (see Fig 10, left). According to the regression model, position is again an important factor in determining syllable duration with the *initial*

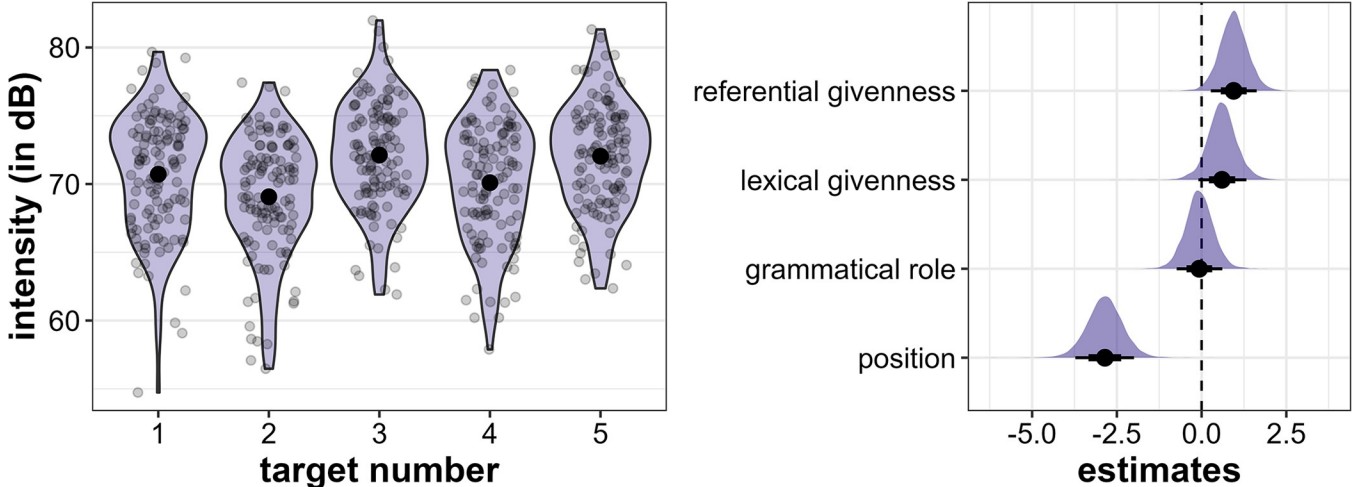

**Fig 10. Results for syllable duration.** Left: Distribution of syllable duration (in ms) across the five target word positions. Right: Posterior estimates for the effects of referential givenness, lexical givenness, grammatical role and position on syllable duration as predicted by the model, means, 66% (thick horizontal lines) and 90% credible intervals (thin lines).

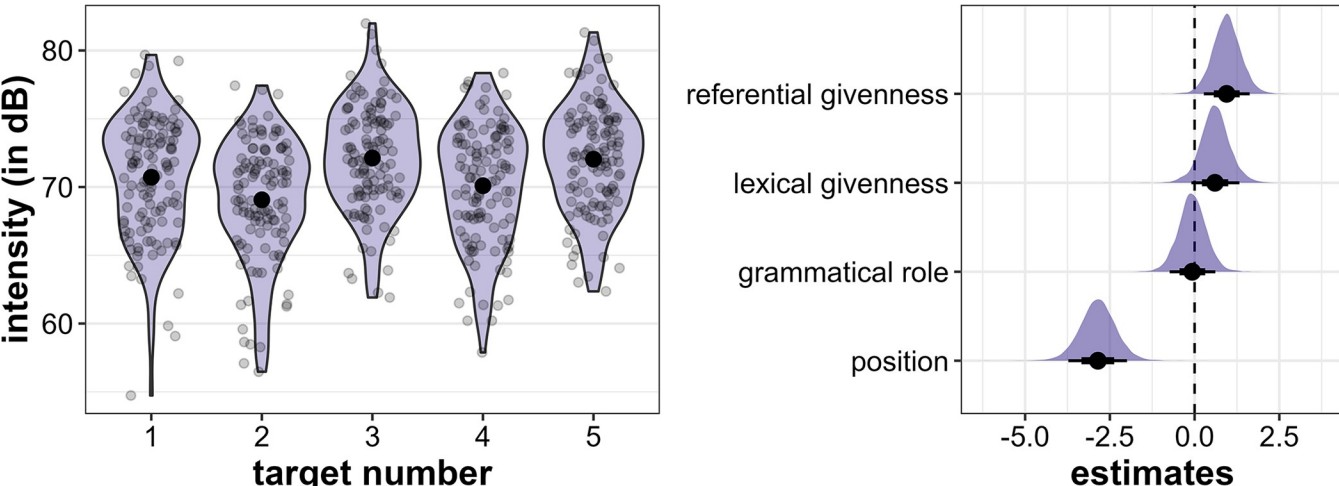

**Fig 11. Results for intensity.** Left: Distribution of intensity (in dB) across the five target word positions. Right: Posterior estimates for the effects of referential givenness, lexical givenness, grammatical role and position on intensity as predicted by the model, means, 66% (thick horizontal lines) and 90% credible intervals (thin lines).

target words reliably exhibiting longer duration (β = -16.12, CI = [-21.6; -10.51], Pr(β>0) = 0.00, Fig 10, right). Recall that we excluded phrase-final target words from the duration analysis due to potential final lengthening effects. That is, the relatively long durations of the initial target words (Target3 and Target5) cannot be explained by the considerable number of phrase breaks following them (see Fig 7). In addition, there is compelling evidence that lexically *new* target words are produced with longer syllable durations (β = 14.89, CI = [5.52; 23.02], Pr(β>0) = 0.99) and also strong evidence that *objects* are longer than *subjects* (β = 5.09, CI = [-0.87; 11.04], Pr(β>0) = 0.92). Although referentially *new* target words also appear to be slightly longer, evidence to support this tendency is very weak (β = 1.44, CI = [-6.08; 8.69], Pr(β>0) = 0.63).

Mean intensity is highest in Target3 (mean = 72.1 dB, sd = 3.9 dB), followed by Target5 (mean = 72.0 dB, sd = 3.9), Target1 (mean = 70.7 dB, sd = 4.4 dB), Target4 (mean = 70.11 dB, sd = 4.3 dB) and Target2 (mean = 69.1 dB, sd = 4.4 dB), but all differences are relatively small (see Fig 11, left). The regression model again supports a strong effect of position: *initial* target words have higher intensity than *medial* words (β = -2.86, CI = [-3.73; -1.99], Pr(β>0) = 0.00, see Fig 11, right). There is also strong evidence that lexically and referentially *new* target words are produced with higher intensity than *given* target words, which is compelling in the case of referential givenness (lexical givenness: β = 0.6, CI = [-0.09; 1.32], Pr(β>0) = 0.93, referential givenness: β = 0.95, CI = [0.27; 1.63], Pr(β>0) = 0.99). Grammatical role does not have a strong effect on intensity (β = -0.07, CI = [-0.74; 0.61], Pr(β>0) = 0.43).

## Discussion

As a general result, and as an answer to Research Question 1, our predicted prominence ranking is widely confirmed by the probabilistic distribution of the target words' accent status. There is only one exception to the prediction, namely that Target5 is produced as prosodically more prominent than Target2, since the former receives a larger proportion of nuclear accents. This leads to the following observed ranking of target words, with decreasing prominence levels: 1 > 4 > 5 > 2 > 3.

The ranking provides evidence in favor of the assumption that the effect of semantic-pragmatic and syntactic cues on a word's prosodic prominence level is to some extent additive, at least when we consider the small selection of factors tested in the present study. In fact, the results for the categorical variable accent status reveal that *all* investigated cues have an effect on the prosodic realization of the target word, with *lexical newness* having the strongest positive effect, followed by *referential newness* and *object* role. Taken together, *lexical newness* has a reliable prominence-enhancing effect on four out of seven prosodic cues (accent status, perceived prominence, TCoG scaling, and syllable duration). *Referential newness* and *object* function also reveal a prominence-enhancing effect, which is less consistent, however. In particular, referential givenness is reliably encoded via accent status and intensity, while grammatical role has a reliable effect on accent status and phrase boundary placement after target words in initial position.

The most consistent and at the same time most unexpected result is the strong prominence-enhancing effect of the *initial sentence position*. That is, an early occurrence in a target sentence pervasively increases a target word's prominence (in five out of seven prosodic cues we find compelling evidence for this effect), which is not the case when it occurs in *medial* position (i.e., as the final argument of the sentence). Sentence-initial target words are not only marked by accents very consistently (hardly any cases of deaccentuation, only one exception with the target word *Parade*, where the article was contrastively accented, see Fig 4) but these accents are nearly always (steeply) rising, i.e., both phonetically and phonologically the most prominent ones. The *initial* position thus constitutes the strongest boosting effect for prosodic prominence in our dataset.

This result is in line with recent findings for German (see, e.g., [39]) as well as with Bolinger's idea of a rhythmical frame for a prototypical English, or rather West Germanic, utterance, i.e., with one prosodic prominence near the beginning of an utterance and one near the end (see State of the art section). It has to be further investigated whether such a rhythmical frame mainly applies to read or generally rather controlled speech or whether it is also consistently found in more spontaneously produced data.

Within the initial position, Target5 is generally more prominent than Target3, particularly in terms of accent status, TCoG scaling and syllable duration. An important reason for the relatively large number of nuclear accents on Target5 as compared to Target3 is the marked syntax of the sentence: The *object*, e.g., *die Praline* in (7), is topicalized or dislocated, so that an intonation phrase (or intermediate phrase) boundary after the target word is favored. As a consequence, the potential prenuclear accent on *Praline* becomes nuclear (as in 42% of all cases in this condition), since it turns into the last accent in the phrase.

(7) **Die Praline (Target5)** wird sie gleich als Vorspeise essen.

('**The chocolate candy** she will eat right away as an appetizer.')

In comparison, only 17% of Target3 tokens, e.g., *die Praline* in (8), receive a nuclear accent, since it functions as a subject and thus occurs in the default position. That is, the manipulation of the grammatical role of the target words in conditions 3 versus 5 (*subject* vs. *object*) results in a difference in the markedness of their position in the sentence, which is reflected in a difference in prosodic phrasing and, in turn, prosodic prominence of the target noun.

(8) **Die Praline (Target3)** war mit Karamell und Nüssen gefüllt.

('**The chocolate candy** was filled with caramel and nuts.')

Let us take a closer look at Research Question 2 and the syntagmatic relation between words in an utterance. We find that the information status of the object in medial position (i.e., target words 1, 2 and 4) has a decisive influence on the strength relation between the last

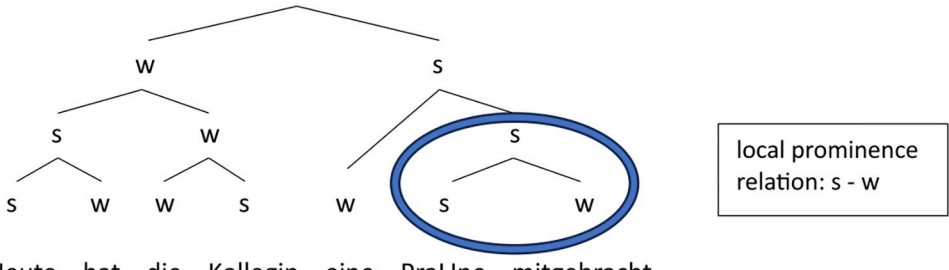

Heute   hat   die   Kollegin   eine   PraLIne   mitgebracht.

(Today, the colleague brought a chocolate candy.)

**Fig 12. Metrical tree displaying the prominence relations in target sentence containing Praline1, where the object Praline ('chocolate candy') carries the nuclear accent (dominated by s-nodes only; underlined).**

argument and the verb as captured by the variable accent status–presumably since it determines the information-structural interpretation of the sentence. There is a clear difference in the percentage of nuclear accents on Target1 (about 96%) and Target2 (about 23%) which stems from the fact that Target1 is both referentially and lexically *new* and Target2 is both referentially and lexically *given*. Thus, we can state that *givenness* (at both levels) has a clearly inhibiting effect on the prominence level of the last object in the phrase. This effect is further enhanced by the relative proximity of the two sentences (separated by one intermediate sentence) and by maintaining the same grammatical role and surface position, which has been found to reduce the level of a word's prosodic prominence (see [66]).

Figs 12 and 13 illustrate the difference in the strength relations between last argument and verb in Target1 and Target2 using examples from the *Praline* story (see (6)). The structural variation is expressed by (simplified) metrical trees which only differ in the relative prominence between the object and the verb at the end of the phrase: The 's' and 'w' nodes (standing for 'stronger' vs. 'weaker') on object and verb are reversed, indicating the abstract (phonological) strength relation between the two elements, which is based on the actual prosodic differences presented in the results section, in particular on the probabilistic distribution of the accent status categories (but also reflected in further parameters such as perceived prominence scores, syllable duration and intensity).

Target4 is referentially *new* but lexically *given*, and this intermediate status between Target1 and Target2 is actually reflected in the produced prominence levels, in particular when looking

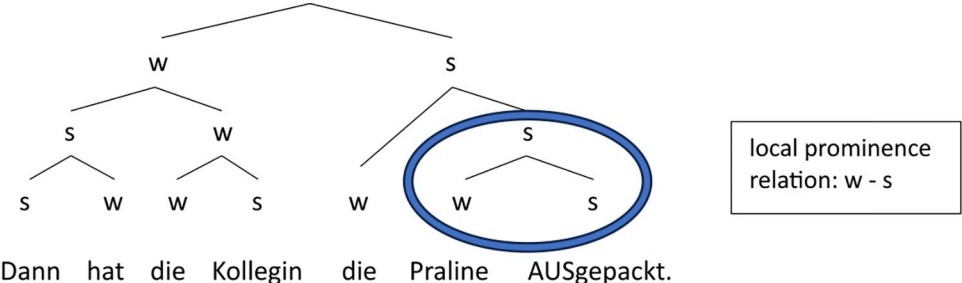

Dann   hat   die   Kollegin   die   Praline   AUSgepackt.

(After that, the colleague unpacked the chocolate candy.)

**Fig 13. Metrical tree displaying the prominence relations in target sentence containing Praline2, where the verbal participle ausgepackt ('unpacked') carries the nuclear accent (dominated by s-nodes only; underlined).**

at the discrete parameters: In comparison with Target2 (referentially *and* lexically *given*), Target4 is marked by more nuclear accents (but also by slightly more deaccentuations) and higher perceived prominence scores (DIMA). These results serve as support for the additive role of the two examined levels of information status on prosodic prominence. In general, however, and as expected, the two conditions turned out to be the least clear-cut in the experiment, since some of the potential prominence cues are in conflict, namely the combination of *object* role (boosting) and (at least one level of) *givenness* (inhibiting). In fact, this conflict was one of the reasons to set up this exploratory study, since no obvious hypothesis for the prosodic realization under such conditions is available. The uncertainty manifests itself in considerable variability within target words 2 and 4, especially in the number of different accent types (Fig 5) as well as broader distributions of perceived prominence levels (Fig 6) and accent status categories (Fig 3; see *Gardine4* in Fig 4 as an example: 60% nuclear accents, 20% prenuclear accents, 20% deaccentuations).

The results for the three conditions containing Target1, Target2 and Target4 show that the *local* prominence relation between object and verb has an important impact on the rest of the utterance, because it determines the position of the nuclear pitch accent, which in turn is decisive for the interpretation of the sentence. The English example (9) may serve as an illustration in analogy to the present dataset (although the word order in English is SVO and in German SOV with the finite verb in second position). Here, a nuclear accent on the verb *reconstructed*, and at the same time deaccentuation of the object *shed* (9a), leads to a coreference reading of *cottage* and *shed*, whereas an accent on *shed* (9b) prevents such an interpretation, i.e., *cottage* and *shed* are different referents. The structure in (9b) is thus equivalent to the condition in which Target1 is *new* and metrically strong (Fig 12), while (9a) is equivalent to the condition in which Target2 is *given* and thus metrically weak (Fig 13).

(9) John has an old cottage.

 a. Last summer he reconSTRUCted the shed.

 b. Last summer he reconstructed the SHED. [67]

While differences in the prominence relations among elements near the end of a sentence are often more fine-grained and the intended accent placement requires some attention on the side of both the speaker and the listener, there is evidence that initial accents are placed less selectively and mainly for rhythmic reasons, as expressed by the concept of the 'rhythmical frame' (as well as Bolinger's *accents of power*). Presumably, the initial position in an utterance is less important for marking differences in information structure, since many referring expressions carry an accent irrespective of being *given* (as in Target3 and Target5; also see [39]). This to some extent 'automatic' placement of accents (which may nevertheless be very prominent, see above) contributes to the metrical structure of an utterance as a whole and may be ascribed to a *global*, overarching layer of prosodic prominence distribution, in contrast to the rather meaning-related local distributions just proposed.

The local prominence relation between object and verb is not only determined by their levels of givenness or potential for a coreference reading but by their semantic weight (see State of the art). Although the word frequency of the verbs in the 'sentence-medial' condition (i.e., Target1, Target2 and Target4) was controlled for in our study, this was not done effectively in terms of their semantic weight, which is evident in the large degree of inter-item variability we observe in our results (see section Discrete prosodic parameters, Fig 4). Some cases in the dataset suggest that the semantic weight of the phrase-final verb plays an important role in a speaker's decision on how to express the prominence relation between the object and the verb. For example, *ausgepackt* (past participle of the verb *auspacken*, 'to unpack') in Fig 13 is presumably

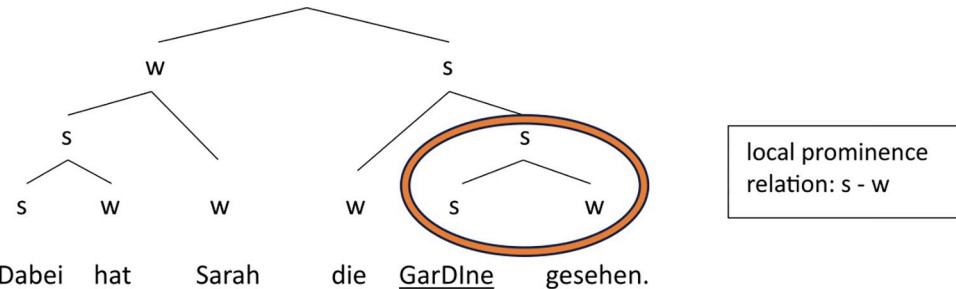

(In the process, Sarah saw the curtain.)

**Fig 14. Metrical tree displaying the prominence relations in target sentence Gardine2, where the object Gardine ('curtain') carries the nuclear accent.**

semantically 'heavier' than *gesehen* (past participle of the verb *sehen*, 'to see') in Fig 14 (displaying the same condition), thus attracting more prosodic prominence and ultimately contributing to drawing the nuclear accent away from the object. That is, the semantic weight of the verb *auspacken* is yet another reason for reversing the prominence relation from stronger-weaker to weaker-stronger, in addition to the *givenness* of the object. In contrast, the object *Gardine* in Fig 14 receives the nuclear accent despite its (referential and lexical) *givenness*—one reason for keeping its strong position in the metrical tree being the fact that the verb *sehen* is semantically light, repelling prosodic prominence. Note that this effect is even stronger in (near) collocations such as *eat* ('essen') and *banana* (*Dabei hat das Mädchen eine Banane (Target4) gegessen*. 'While doing so, the girl was eating a banana.'). Here, the semantic weight of the verb is very low, leading to 100% nuclear accents on the object (see Fig 4).

In sum, Bolinger's [24] observation cited in the State of the art section proves largely true in one of the central results of our study: What counts in order to appropriately reflect the semantic-pragmatic meaning of an utterance is the *relative* prosodic prominence between the elements, in combination with syntactic cues—which in turn determines the position of the nuclear accent.

Finally, a note on speaking style, gender, and especially focus is necessary. None of the participants in our study used an emotionally marked or particularly involved speaking style that would dramatically increase the level of prosodic prominence. Although the task was listener-oriented in that it elicited clear and comprehensible speech from the participants, the outcome rather resembled an active and engaged reading style. Gender did not play a role, although we have to state that the dataset is not balanced across gender (see section Participants and experimental procedure).

The role of focus is somewhat more complicated. If we accept the assumption that focus expresses the pragmatically most important part of an utterance, we also have to accept that focus is orthogonal to other levels of information structure, such as information status. That is, it can overwrite givenness, especially in contrastive contexts. As such, focus is crucial for the prosodic realization of utterances. Nevertheless, we refrained from including focus as a variable in the present dataset, for several reasons: First, contrastive focus was not purposefully elicited. Second, focus is a less objective and less easily operationalizable category than others that we examined in our data. For example, the widely used method of *Question-Answer Congruence* (see, e.g., [68]) is not available in connected texts like ours, and also the promising *Question under Discussion* (QUD) method for the information-structural analysis of naturalistic texts, in which implicit questions are identified (see [69]), is too strict in one important point: The *Maximize-Q-Anaphoricity* principle requires that QUDs should contain as much

(referentially or lexically) *given* material as possible, which in turn cannot be part of the focus, which functions as the answer to the QUD. Only for focus and contrastive topics (which are part of the background), prosodic highlighting by pitch accents is expected. For our data, and this is the third reason why focus is not included as a variable here, this means that only Target1 would be classified as focused while all other target words would be in the background. Thus, the distribution of *background vs. focus* would coincide with *lexically given vs. lexically new*, making an extra variable 'focus' obsolete. Interestingly, the variable *lexical newness* proved to be a highly consistent prominence-boosting factor in our data, suggesting a certain connection with the (potential) focus level after all.

Generally, if no reliable classification of the focus structure is available that is *independent from its prosodic realization*, an analysis is in danger of circularity, for example when extrapolating from a pitch accent to the presence of focus, along the lines of the strict *Focus-to-Accent* approach proposed by [70]. An appropriate detour seems to be an investigation of the probabilistic distribution of pitch accents (and other, gradient, prosodic cues) in different sentence structures including the meaning-related factors that might be responsible for the prosodic differences observed. This is the approach we took in the present study.

As a next step, the outcome of our production study will be verified and further investigated in a prominence rating task with gradual visual analogue scales (from 0–100) for each word. The prominence ratings can be used as the dependent variable for a *Random Forest* analysis, which provides information on which prosodic, syntactic or pragmatic factor is more important, and, as a consequence, has to be weighted as stronger. It will be interesting to see, for instance, whether rising but prenuclear accents on initial target words will perceptually outrank falling accents in nuclear position, and to what extent the judgments are influenced by the level of a referent's givenness. The result of this analysis is thus a more fine-grained ranking of factors relevant for the perception of prominence and a step towards the development of a more comprehensive metrical model of discourse meaning.

## Conclusions

The results of our exploratory production study revealed that among the examined semantic-pragmatic and syntactic variables *lexical newness* and, contrary to our predictions, *initial sentence position* had the strongest boosting effect on the target words' prosodic prominence. Furthermore, the influence of all four factors, i.e., including *referential newness* and *object* role, had a weak additive effect, which was particularly evident in the probabilistic distribution of the target words' accent status. Our predicted prominence ranking was broadly confirmed for this prosodic parameter.

Nearly all prosodic cues we investigated, both discrete and gradient, were employed in the expression of prominence triggered by the controlled semantic-pragmatic and syntactic variables. Some cues were more clearly affected by meaning-related differences than others, in particular accent status, but also TCoG scaling or syllable duration showed an effect of more than one variable. Especially the level of (both referential and lexical) givenness turned out to be crucial for a speaker's decision to produce an appropriate prosodic prominence relation between an object and a verb at the end of a sentence. However, in addition to the level of givenness the relative semantic weight of both object and verb also played a role in determining the position of the nuclear accent in order to get the intended meaning of an utterance across.

## Acknowledgments

We thank Tiffany Zukas for creating the pictures used in the reading task and her help with data elicitation and annotation. In addition, we are grateful to our 'confederates' Timo

Buchholz, Tobias Edele, Alicia Janz, Anne Lützeler, Lena Pagel, Christine Röhr, Tobias Schröer, Malin Spaniol, Simon Wehrle, Katinka Wüllner and Barbara Zeyer.

## Author Contributions

**Conceptualization:** Stefan Baumann, Janne Lorenzen.

**Data curation:** Stefan Baumann, Janne Lorenzen.

**Formal analysis:** Janne Lorenzen.

**Funding acquisition:** Stefan Baumann.

**Investigation:** Stefan Baumann, Janne Lorenzen.

**Methodology:** Stefan Baumann, Janne Lorenzen.

**Project administration:** Stefan Baumann.

**Supervision:** Stefan Baumann.

**Writing – original draft:** Stefan Baumann.

**Writing – review & editing:** Stefan Baumann, Janne Lorenzen.

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
