## [Decision Letter · Decision Letter 0]

26 Dec 2023

PONE-D-23-34944Boosting or inhibiting - How semantic-pragmatic and syntactic cues affect prosodic prominence relations in GermanPLOS ONE

Dear Dr. Baumann,

Thank you for submitting your manuscript to PLOS ONE. After careful consideration, we feel that it has merit but does not fully meet PLOS ONE’s publication criteria as it currently stands. Therefore, we invite you to submit a revised version of the manuscript that addresses the points raised during the review process.

We look forward to receiving your revised manuscript.

Kind regards,

John Blake, PhD

Academic Editor

PLOS ONE

Journal Requirements:

"This work was supported by the German Research Foundation (DFG) as part of the SFB1252 Prominence in Language (Project-ID 281511265), project A07 Metrical prominence – Scales and structures."

Additional Editor Comments:

Reviewer 1 has made a few minor suggestions to improve your manuscript, all of which are relatively easy to implement.

I contacted multiple potential reviewers for your manuscript and the responses were either that there was a conflict of interest (i.e. they were known to you), your paper was too specialized, or that having read the paper they felt they were unable to offer any suggestions and so could not submit an actionable review. I spoke directly to Reviewer 1 to discuss the content of your paper, and on the basis of the review, the subsequent discussion and the fact that the non-submitting reviewers had no suggestions, I am happy to proceed with just one review. 

Reviewers' comments:

Reviewer's Responses to Questions

**Comments to the Author**

1. Is the manuscript technically sound, and do the data support the conclusions?

Reviewer #1: Yes

2. Has the statistical analysis been performed appropriately and rigorously? 

Reviewer #1: Yes

3. Have the authors made all data underlying the findings in their manuscript fully available?

Reviewer #1: Yes

4. Is the manuscript presented in an intelligible fashion and written in standard English?

Reviewer #1: Yes

5. Review Comments to the Author

Reviewer #1: Introduction:

The Introduction section is well written and the research question is described clearly. The previous literature is reviewed such that a clear picture of the research question emerges.

Section 2.2:

It is mentioned that participants "originat[e] from five different federal states of Germany." Are there any potential differences between dialects of the participants? If not, state this explicitly.

Is there any possible effect of gender? With only 4 male and 11 female speakers, the data is not balanced across gender. If there are differences, it would not necessarily be clear then due to this imbalance. This should be mentioned explicitly as a limitation.

Section 2.3:

Include the inter-rater reliability between the two annotators for word prominence level annotation. Also, how many words required a consensus from a third expert?

When normalizing F0 by gender, what was the justification to use 75 Hz for males and 120 Hz for females? Why not normalize by individual mean F0 (per participant) instead to account for participant differences (regardless of gender)?

Section 2.4: The statistical analysis is nicely presented.

Section 4: The authors’ discussion following example (9) offers a plausible answer to the question of why initial position tends to attract nuclear accent: It’s a global preference and doesn’t depend on anything else. The authors might consider reframing their initial predictions then, or at least entertaining this possibility from the outset in the Introduction section rather than introducing it for the first time in the Discussion section. It need not be presented as such a surprising result, in other words.

6. PLOS authors have the option to publish the peer review history of their article (what does this mean?). If published, this will include your full peer review and any attached files.

Reviewer #1: No

---

## [Author Response · Author response to Decision Letter 0]

22 Jan 2024

Dear Dr. Blake,

We are happy to submit our revision for the manuscript entitled “Boosting or inhibiting - How semantic-pragmatic and syntactic cues affect prosodic prominence relations in German”, authored by Stefan Baumann (corresponding author), and Janne Lorenzen.

We thank you and the reviewer for the very helpful feedback. We have considered the points raised and give a point-by-point description below (our answers are given in italics).

Furthermore, please change our online submission form on our behalf by adding to the statement on financial disclosure ("This work was supported by the German Research Foundation (DFG) as part of the SFB1252 Prominence in Language (Project-ID 281511265), project A07 Metrical prominence – Scales and structures.") the following sentence: "The funders had no role in study design, data collection and analysis, decision to publish, or preparation of the manuscript."

Finally, we included our full ethics statement in the "Participants and experimental procedure" section of our manuscript file, including the full name of the ethics committee who approved our study.

With kind regards,

Stefan Baumann and Janne Lorenzen

Response to Reviewer 1

Introduction:

The Introduction section is well written and the research question is described clearly. The previous literature is reviewed such that a clear picture of the research question emerges.

Section 2.2:

It is mentioned that participants "originat[e] from five different federal states of Germany." Are there any potential differences between dialects of the participants? If not, state this explicitly.

Thank you for this question. We added the statement "None of them spoke in a non-standard variety" to clarify this point.

Is there any possible effect of gender? With only 4 male and 11 female speakers, the data is not balanced across gender. If there are differences, it would not necessarily be clear then due to this imbalance. This should be mentioned explicitly as a limitation.

We added this information to a paragraph in the Discussion section, when talking about speaking style and focus. In particular, the following sentence was included: "Gender did not play a role, although we have to state that the dataset is not balanced across gender (see section Participants and experimental procedure)." 

Section 2.3:

Include the inter-rater reliability between the two annotators for word prominence level annotation. Also, how many words required a consensus from a third expert?

We included the inter-rater agreement for all prosodic annotations (phrase boundary, DIMA tone level, GToBI accent types and perceived prominence). The inverse of the agreement rates constitute the amount of annotations requiring a consensus with the third expert (e.g. an agreement rate of 94% means that we had to decide on a consensus annotation in 6% of the cases).

When normalizing F0 by gender, what was the justification to use 75 Hz for males and 120 Hz for females? Why not normalize by individual mean F0 (per participant) instead to account for participant differences (regardless of gender)?

We agree that a participant-specific normalization is the more thorough process and have adjusted the TCoG scaling variable and its analysis in this regard. The results do not change following this procedure.

Section 2.4: The statistical analysis is nicely presented.

Thank you. :-)

Section 4: The authors’ discussion following example (9) offers a plausible answer to the question of why initial position tends to attract nuclear accent: It’s a global preference and doesn’t depend on anything else. The authors might consider reframing their initial predictions then, or at least entertaining this possibility from the outset in the Introduction section rather than introducing it for the first time in the Discussion section. It need not be presented as such a surprising result, in other words.

First of all, initial position generally attracts accents, but mainly prenuclear ones (not nuclear accents). 

Secondly, we still want to say that we did not predict the initial position to be that strong, as we want to be transparent about our initial assumptions, but we tried to attenuate the "surprisal" to some extent, as suggested, by changing the wording, e.g. to "contrary to our predictions" (Conclusions section). We also prepared the rhythmical frame idea in the Introduction a bit more. In the Abstract, we deleted "somewhat surprisingly" completely.

Thank you again for your valuable feedback!

---

## [Decision Letter · Decision Letter 1]

16 Feb 2024

Boosting or inhibiting - How semantic-pragmatic and syntactic cues affect prosodic prominence relations in German

PONE-D-23-34944R1

Dear Dr. Baumann,

We’re pleased to inform you that your manuscript has been judged scientifically suitable for publication and will be formally accepted for publication once it meets all outstanding technical requirements.

Kind regards,

John Blake, PhD

Academic Editor

PLOS ONE

Reviewers' comments:

Reviewer's Responses to Questions

**Comments to the Author**

1. If the authors have adequately addressed your comments raised in a previous round of review and you feel that this manuscript is now acceptable for publication, you may indicate that here to bypass the “Comments to the Author” section, enter your conflict of interest statement in the “Confidential to Editor” section, and submit your "Accept" recommendation.

Reviewer #1: All comments have been addressed

2. Is the manuscript technically sound, and do the data support the conclusions?

Reviewer #1: Yes

3. Has the statistical analysis been performed appropriately and rigorously? 

Reviewer #1: Yes

4. Have the authors made all data underlying the findings in their manuscript fully available?

Reviewer #1: Yes

5. Is the manuscript presented in an intelligible fashion and written in standard English?

Reviewer #1: Yes

6. Review Comments to the Author

Reviewer #1: The revisions took into account all the comments satisfactorily. The agreement rate for annotation of accentual vs. non-accentual tones (71%) was slightly low for a binary choice rating and may be worth commenting on in the final version. However, the 3rd party consensus is enough to catch most/all of these.

7. PLOS authors have the option to publish the peer review history of their article (what does this mean?). If published, this will include your full peer review and any attached files.

Reviewer #1: No

---

## [Editor Report · Acceptance letter]

27 Feb 2024

PONE-D-23-34944R1 

PLOS ONE

Dear Dr. Baumann, 

I'm pleased to inform you that your manuscript has been deemed suitable for publication in PLOS ONE. Congratulations! Your manuscript is now being handed over to our production team.

Kind regards, 

on behalf of

Dr. John Blake 

Academic Editor

PLOS ONE